# Evaluation of Artificial Precipitation Enhancement Using UNET-GRU Algorithm for Rainfall Estimation

Renfeng Liu [1], Huabing Zhou [2], Dejun Li [3,*], Liping Zeng [4] and Peihua Xu [3]

1   School of Mathematics and Computer Science, Wuhan Polytechnic University, Wuhan 430023, China;
    claye.liu@whpu.edu.cn
2   Hubei Key Laboratory of Intelligent Robot, Wuhan Institute of Technology, Wuhan 430205, China
3   Hubei Meteorological Service Center, Wuhan 430205, China
4   Guizhou Meteorological Service Center, Guiyang 550081, China
*   Correspondence: esldj@163.com

**Abstract:** The evaluation of the effects of artificial precipitation enhancement remains one of the most important and challenging issues in the fields of meteorology. Rainfall is the most important evaluation metric for artificial precipitation enhancement, which is mainly achieved through physics-based models that simulate physical phenomena and data-driven statistical models. The series of effect evaluation methods requires the selection of a comparison area for effect comparison, and idealized assumptions and simplifications have been made for the actual cloud precipitation process, leading to unreliable quantitative evaluation results of artificial precipitation effects. This paper proposes a deep learning-based method (UNET-GRU) to quantitatively evaluate the effect of artificial rainfall. By comparing the residual values obtained from inverting the natural evolution grid rainfall of the same area under the same artificial rainfall conditions with the actual rainfall amount after artificial rainfall operations, the effect of artificial rainfall can be quantitatively evaluated, effectively solving the problem of quantitative evaluation of artificial precipitation effects. Wuhan and Shiyan in China are selected to represent typical plains and mountainous areas, respectively, and the method is evaluated using 6-min resolution radar weather data from 2017 to 2020. During the experiment, we utilized the UNET-GRU algorithm and developed separate algorithms for comparison against common persistent baselines (i.e., the next-time data of the training data). The prediction of mean squared error (*MSE*) for these three algorithms was significantly lower than that of the baseline data. Moreover, the indicators for these algorithms were excellent, further demonstrating their efficacy. In addition, the residual results of the estimated 7-h grid rainfall were compared with the actual recorded rainfall to evaluate the effectiveness of artificial precipitation. The results showed that the estimated rainfall was consistent with the recorded precipitation for that year, indicating that deep learning methods can be successfully used to evaluate the impact of artificial precipitation. The results demonstrate that this method improves the accuracy of effect evaluation and enhances the generalization ability of the evaluation scheme.

**Keywords:** evaluation of artificial precipitation enhancement (EoAPE); UNET-GRU; rainfall estimation



## 1. Introduction

Currently, artificial precipitation enhancement is an important approach for alleviating drought conditions and increasing water storage in many countries and regions. However, the evaluation method for the catalytic effect of cloud seeding has been questioned on occasion [1–5]. The evaluation method for the effect of artificial precipitation enhancement mainly focuses on the following four aspects:

(1)   Comparing the target cloud operation with contrast cloud radar echoes, satellite inversion products, etc., and conducting physical inspections [6–11];
(2)   Statistical evaluation of rainfall in the affected area and comparison area [12–17];



(3)     Adopting a comprehensive inspection method that combines numerical model simulation and live observation [18–21];

(4)     Random evaluation of rainfall increase rates [22–26].

In (1) and (2), due to the significant spatiotemporal variability of precipitation in China, it is challenging to identify an appropriate reference area for comparison. For (3), live observation data are usually collected at discrete stations with 1-h resolution, while the artificial rainfall process usually lasts for only 4–7 h or even shorter. Thus, it is difficult to distinguish the increase in precipitation due to artificial means from natural variation using limited data. As for (4), the randomization method requires a sufficiently long experiment period to test the effect of single or a few artificial precipitation enhancement operations. However, this method wastes half of the operation time and may not completely eliminate human influence, making non-randomization methods more commonly used in practice.

Deep learning, a promising branch of machine learning, has received extensive attention in recent years and has been widely used in pattern recognition, image processing, fault detection, classification, and prediction tasks [27–35]. Compared to shallow models, deep learning has three major characteristics: unsupervised feature learning, strong generalization ability, and big data training. Deep learning-based predictive models are generally accepted to exhibit attractive performance in terms of accuracy, stability, and effectiveness. As rainfall is a direct indicator to evaluate the effect of artificial precipitation, it is important to compare the accuracy of natural precipitation variation in the last 4–7 h under climatic conditions within half an hour before the implementation of artificial precipitation. Therefore, short-term prediction algorithms based on deep learning can be used to evaluate the effect of artificial precipitation enhancement. A local prediction system based on deep learning was proposed by [36] to predict short-term rainfall at 16 points in Japan individually, using various forms of massive data provided by the Japan Meteorological Agency (JMA). However, as there are only a few artificial precipitation enhancement operations in each region per year, data is limited. Work in [37] proposed a SmaAT-UNet architecture for short-term forecasts, which is applicable to cases with small amounts of data and calculation. However, the predicted precipitation maps of all models are quite blurry due to the use of *MSE* as the guiding loss function, which is biased towards blurry images.

In summary, while deep learning-based prediction algorithms have been widely applied in the meteorological field, they require a large and diverse amount of training data. For the evaluation of artificial precipitation enhancement, deep learning algorithms have faced challenges due to the scarcity and uneven distribution of training samples, which limits their predictive accuracy and impacts the credibility of the evaluation results. As a result, traditional methods based on physical models or statistical principles remain the mainstream approach. However, these methods often require the selection of a comparison area (as shown in Figure 1) and can only provide qualitative evaluations, which may not lead to convincing conclusions. Therefore, we propose a deep learning algorithm based on small sample training to invert the 1–7 h rainfall natural catalysis situation in the same area that meets the meteorological conditions of artificial precipitation enhancement. We evaluate the effectiveness of artificial precipitation enhancement by comparing the residual between the natural catalysis rainfall and the actual rainfall enhancement. The grid-based residual data can be verified through the collected data from rainfall monitoring stations, thus proving the feasibility of our evaluation plan.

This paper is organized as follows. In Section 2, we provide a brief overview of related research on evaluating the effect of artificial precipitation enhancement. Section 3 describes the proposed UNET-GRU architecture and other models used for effect evaluation. In Section 4, we present the experiments conducted and the results obtained. A discussion of the results is provided in Section 5. Finally, we conclude with some remarks in Section 6.

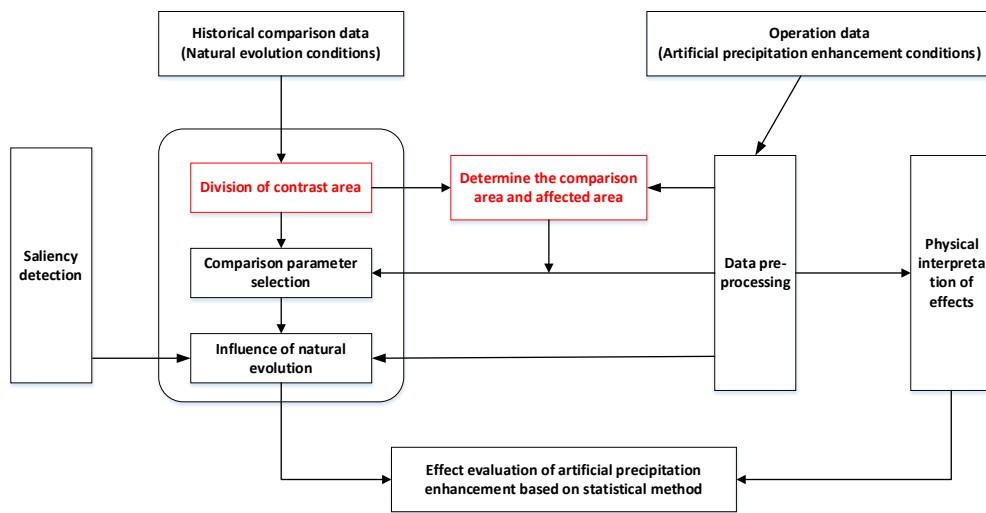

**Figure 1.** The classical evaluation method of artificial precipitation enhancement effect.

## 2. Related Work

### 2.1. Classical Evaluation Methods Model in Meteorological Field

The traditional approach for evaluating the effect of artificial precipitation enhancement relies on statistical methods (Figure 1). This involves determining the impact area and comparison area of the operation, and estimating the difference between natural precipitation and measured precipitation based on these two regions to determine the effect of the operation. However, in many cases, the impact area and comparison area of artificial precipitation enhancement are not fixed. In such cases, the floating comparison area method is usually employed to evaluate the effect, which involves dividing the comparison area based on principles such as similar weather systems, similar terrain, and good precipitation correlation. Therefore, a reasonable division of the comparison area is a prerequisite for the statistical evaluation of the effect of artificial precipitation enhancement.

### 2.2. Application of Deep Learning Model in Meteorological Field

Deep learning methods, including convolutional neural networks (CNNs) and recurrent neural networks (RNNs), have been widely applied in weather forecasting. For instance, Kevin Trebing et al. introduced the SMART-UNET algorithm with CBAM attention block for short-term precipitation forecasting [37]. Xingjian Shi et al. utilized radar CAPPI data to achieve regional precipitation forecasting with convolutional LSTM [38]. Jing J et al. introduced the MLC-LSTM algorithm to extrapolate echo sequences by leveraging the space time correlation between multi-level weather radar echoes [39]. The deep learning methods can handle the nonlinear problem of precipitation and radar echo that cannot be solved by statistical methods and numerical simulations. Furthermore, they can better simulate the natural evolution of natural precipitation and radar echo. However, these methods have rarely been used in evaluating the effect of artificial precipitation enhancement. The evaluation of this effect requires predicting the rainfall under natural conditions, which can also be achieved through deep learning methods. Therefore, introducing deep learning techniques to improve the existing evaluation methods of artificial precipitation enhancement can have significant theoretical and practical value.

### 2.3. UNET

In the field of computer vision, the FCN is a well-known image segmentation network. In the field of medical image processing, UNET is a popular network for semantic segmentation tasks, often used as a baseline [40]. The UNET architecture consists of two parts: feature extraction and upsampling. This structure is commonly referred to as an encoder decoder structure. The network resembles a letter U, hence its name. As shown in Figure 2, the input image is first convolved and pooled. In the original UNET paper, the

image is pooled four times, resulting in features of sizes $144 \times 144$, $72 \times 72$, $36 \times 36$, and $18 \times 18$. The $18 \times 18$ feature map is then upsampled to obtain a $36 \times 36$ feature map, which is concatenated with the previous $36 \times 36$ feature map to preserve channel information. The concatenated feature map is then convolved and upsampled to obtain a $72 \times 72$ feature map, which is again concatenated with the previous $72 \times 72$ feature map. After four rounds of upsampling, a $288 \times 288$ prediction result with the same size as the input image can be obtained.

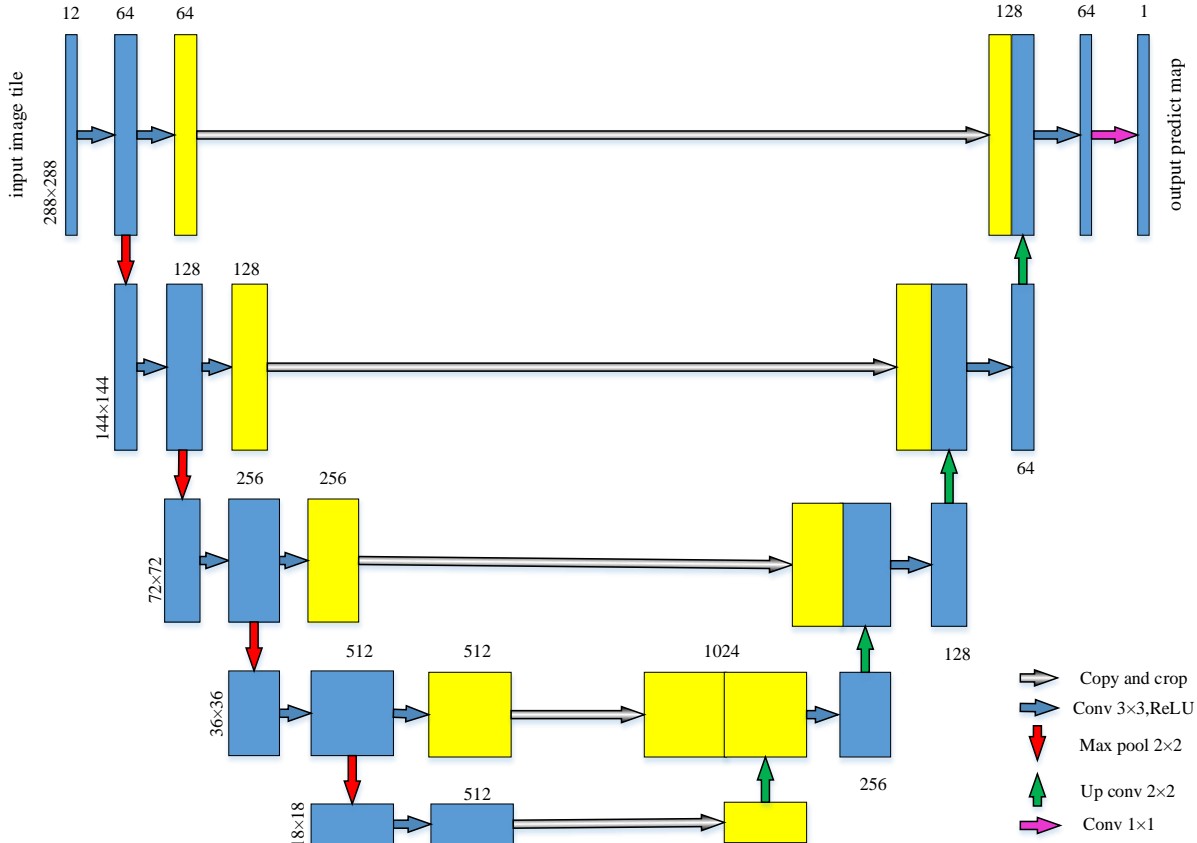

**Figure 2.** UNET structure.

UNET has several advantages. Firstly, as the network layer becomes deeper, the feature map's field of view increases. While shallow convolution focuses on texture features, deep networks focus on essential features, making both deep and shallow features grid meaningful. Secondly, larger-sized feature map edges obtained through deconvolution are often lack of information. This is because, during downsampling, some edge features are lost, and these cannot be retrieved during upsampling. Therefore, one edge feature can be retrieved through feature splicing.

UNET has found wide applications in medical image segmentation due to four characteristics of medical images. Firstly, medical images have simple semantics and fixed positions, making their semantic information relatively simple, and there is no need to filter out useless information. All features of medical images are important, including both low-level features and high-level semantic features, making the skip connection structure (feature splicing) of the U-shaped structure better used. Secondly, medical images are difficult to obtain, and the dataset available is usually very small. The amount of data may only be a few hundred, or even fewer than a hundred. Therefore, it is easy to over-fit large networks like DeepLabv3+. While large networks have stronger image representation ability, simpler and fewer medical images do not have much content to express. As a result, some people find that, in the decimal scale, the segmented SOTA model has no advantage over the lightweight UNET. Thirdly, medical images are often multimodal, such as in the

ISLES, data on modalities such as CBF, MTT, and CBV can usually be obtained. In medical imaging tasks, we often need to design our network to extract different modal features, lightweight and simple UNET more operation space.

In the field of artificial precipitation enhancement effect inspection, the UNET algorithm has been found to be applicable for short-term rainfall prediction in the same region, because it has the ability to handle the above three characteristics. Firstly, the area of interest for precipitation enhancement (artificial rainfall area) is usually specific and localized (characteristic 1). Secondly, the number of opportunities for artificial precipitation enhancement in a specific area is limited each year (characteristic 2), with generally only tens of operations per year in the same area, each operation lasting for 4–7 h. Finally, the available data for precipitation enhancement evaluation include various modal data formats such as radar data (MCR, MTOP, CAPPI, etc.), satellite data, and real-time data (characteristic 3). Therefore, the UNET algorithm can be used to evaluate the effect of artificial precipitation enhancement.

## 3. Methods

### 3.1. Effect Evaluation Method Based on Deep Learning

The classical method involves determining a contrast area and an affected area, and comparing the rainfall in these two areas. However, the comparison and affected areas cannot be entirely consistent, which may result in less convincing evaluation results. To address this, we propose setting the affected area as the comparison area to ensure that both areas have the same meteorological conditions (e.g., wind speed, air temperature, and radar data) prior to the implementation of artificial precipitation enhancement. By inverting the deduction process under natural conditions (4–7 h), we can use it as a comparison area to compare with the affected area. Thus, we transform the evaluation method into Figure 3.

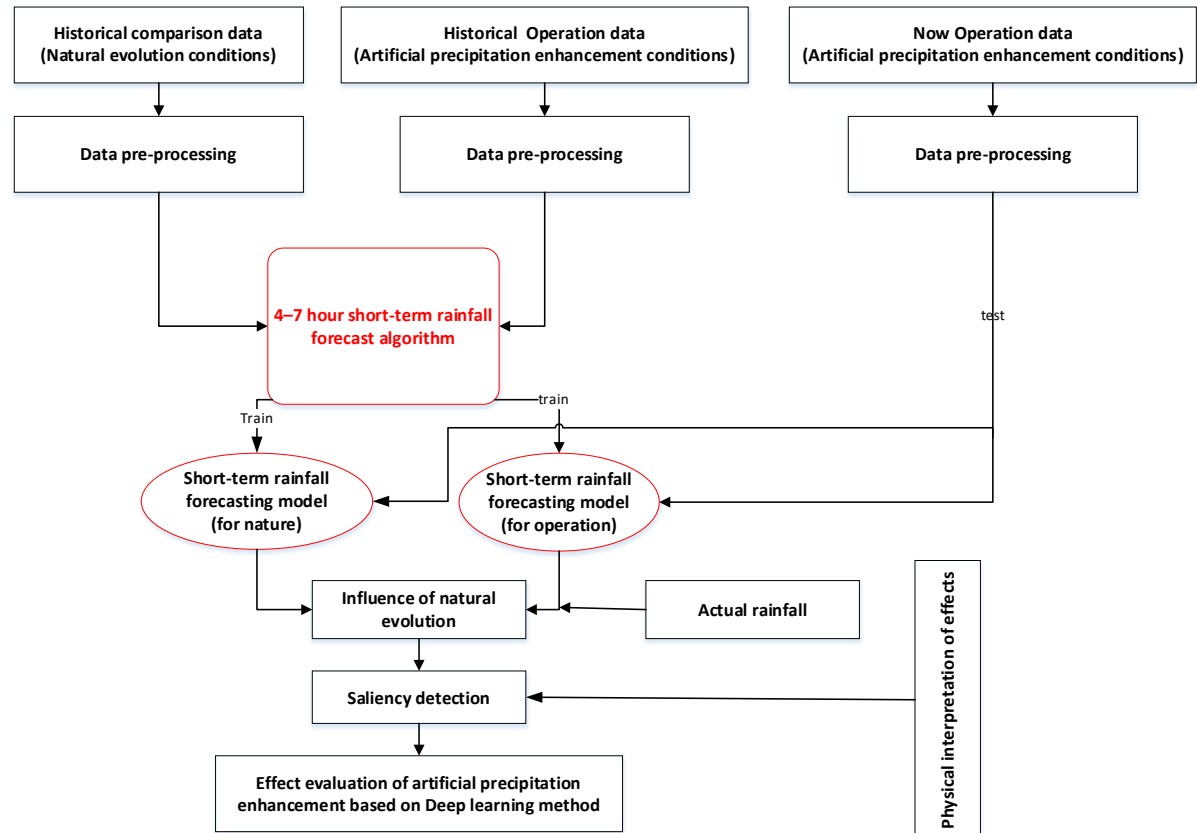

**Figure 3.** Evaluation scheme of artificial precipitation enhancement based on deep learning.

This paper proposes an evaluation method for artificial precipitation enhancement effects based on deep learning. The method focuses on designing a reasonable short-term rainfall forecasting algorithm and building a training model for natural evolution and artificial precipitation enhancement conditions separately using historical data. After implementing artificial precipitation enhancement in the region, the current meteorological data can be inputted into both models to obtain rainfall predictions under natural evolution and artificial precipitation enhancement conditions, allowing for the evaluation of artificial precipitation enhancement effects. To verify the model's applicability, the predicted rainfall from the artificial precipitation enhancement model can be compared with actual rainfall. It should be noted that the general rainfall forecast model is not applicable in this method due to the small dataset under artificial precipitation enhancement conditions (dozens of datasets per year). Deep learning algorithms usually require tens of thousands of datasets for training, which can lead to poor model generalization ability or even overfitting. Therefore, it is necessary to propose a short-term rainfall forecasting algorithm that supports small dataset training.

### 3.2. UNET-GRU Algorithm

Therefore, we propose a convolution GRU algorithm that integrates UNET, as shown in Figure 4, for short-term rainfall forecasting. The algorithm utilizes the four modes of weather radar data with 6-min resolution (MTOP, MCR, MVIL, and CAPPI) to generate an image, which is then processed by the UNET network to extract modal characteristics. As different modes contain dense links, we introduce convolution GRU to better utilize the relationship between continuous radar data and explain the nonlinearity in multimodal data modeling. The output of the model is the grid rainfall at a specific time in a certain area. Due to the large data volume of CAPPI data with its 24 layers, the training sequence is not included in this experiment, as our laboratory computer is not capable of handling such large datasets. Nonetheless, the proposed algorithm still yields satisfactory results.

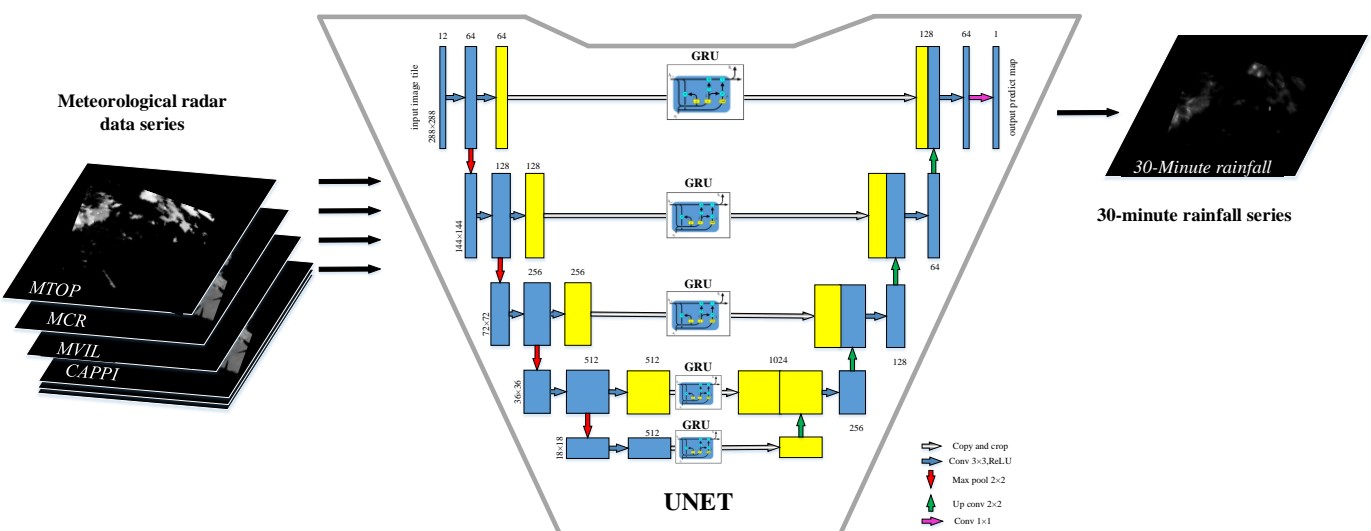

**Figure 4.** UNET-GRU algorithm block diagram.

GRU (as shown in Figure 5) is a highly effective variation of the LSTM network, featuring a simpler structure with comparable performance. As a result, it has become a popular choice in many applications. In common with LSTM, GRU is capable of addressing the long dependency problem in RNNs. The GRU model has only two gates: a reset gate and an update gate. The reset gate functions similarly to LSTM's forgetting gate, but it does

not forget the information of the memory unit $C_{t-1}$ from the previous time step. Instead, it forgets the information of the hidden layer unit $h_{t-1}$ from the previous time step:

$$r_t = \sigma(W_r \cdot [h_{t-1}, x_t] + b_r) \tag{1}$$

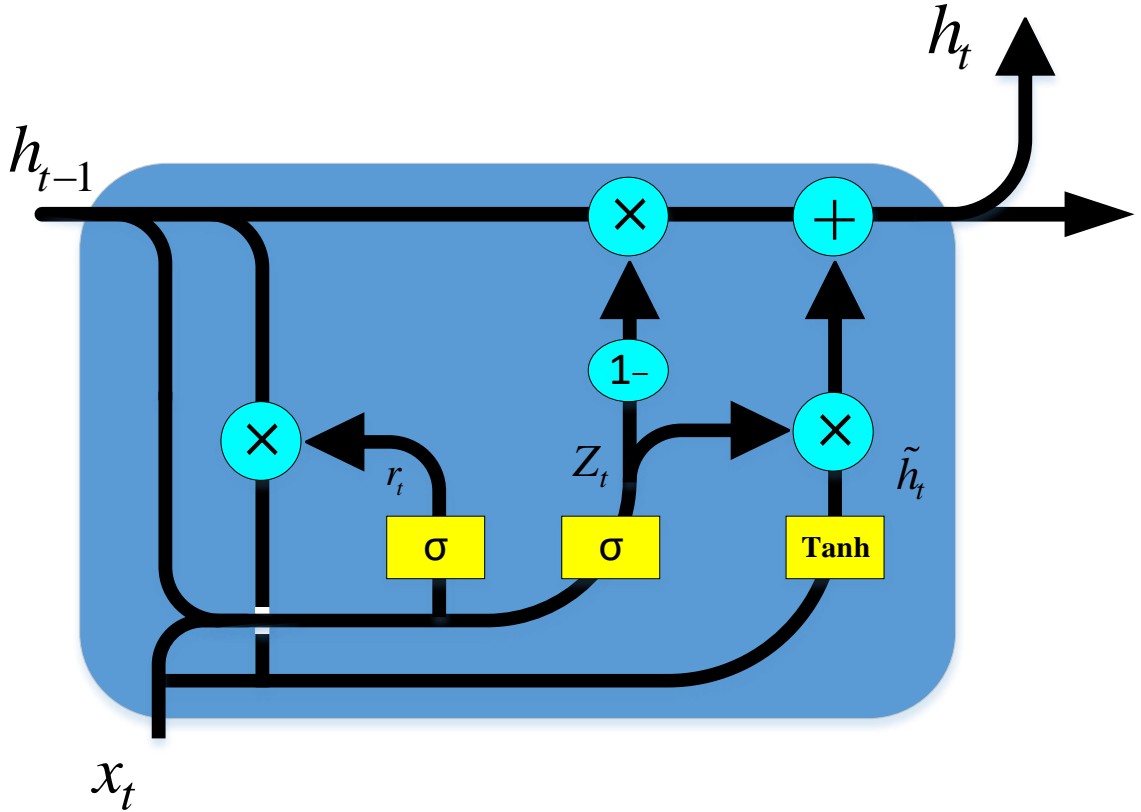

**Figure 5.** GRU algorithm block diagram.

The function of update gate is different from that of LSTM. It controls the balance between the hidden layer state $h_{t-1}$ at the previous moment and the current input information.

$$Z_t = \sigma(W_z \cdot [h_{t-1}, x_t] + b_z) \tag{2}$$

The input information is $r_t \cdot h_{t-1}$ after forgetting,

$$\widetilde{h}_t = \tanh(W_h \cdot [r_t \cdot h_{t-1}, x_t] + b_h) \tag{3}$$

$h_t$ after balance:

$$h_t = (1 - Z_t) \cdot h_{t-1} + z_t \cdot \widetilde{h}_t \tag{4}$$

[] represents concat, · represents element-level multiplication.

### 3.3. Other Models

As a comparison, we trained both the UNET and CoGRU (GRU algorithm supporting convolution operation) architectures, resulting in a total of three models compared in this study. Table 1 presents a comparison of the models' parameters. When examining the standard UNET architecture and our proposed UNET-GRU architecture, it is evident that the latter has slightly more parameters—approximately 21.6 million compared to approximately 17.3 million.

**Table 1.** Number of parameters of the compared models.

| Model | Parameters |
|---|---|
| UNET | 17,272,577 |
| CoGRU 2 | 210,701 |
| UNET-GRU | 21,555,009 |

*3.4. Training*

All three models mentioned above were trained for a maximum of 200 epochs. An early stopping criterion was utilized, which stopped the training process if the validation loss did not improve in the last 15 epochs. This criterion was satisfied in all training iterations, so the maximum of 200 epochs was never reached. Additionally, a learning rate scheduler was employed, which decreased the learning rate to one-tenth of the previous rate when the validation loss did not improve for four epochs. The initial learning rate was set to 0.001, and the Adam [36] optimizer with default values was used. The training was conducted on a single NVIDIA GeForce RTX 3090 Super with 24 GB of VRAM.

*3.5. Model Evaluation*

The loss function used in this study is the mean squared error (*MSE*) between the output images and the ground truth images. The *MSE* is calculated as follows:

$$MSE = \frac{\sum_{i=1}^{n}(y_i - \hat{y}_i)^2}{n} \tag{5}$$

where $n$ represents the number of samples, $y_i$ is the value of the ground truth, and $\hat{y}_i$ is the value of the prediction. Apart from mean squared error (*MSE*), various performance evaluation metrics are calculated, including precision, recall (probability of detection), accuracy, F1-score, critical success index (*CSI*), false alarm rate (*FAR*), and Heidke skill score (*HSS*). For the precipitation map dataset, these metrics are computed for rainfall greater than a threshold of 0.5 mm/h. To achieve this, a Boolean mask is generated for each pixel of the predicted output and target images using the specified threshold. Based on this, the number of true positives (*TP*) (prediction = 1, target = 1), false positives (*FP*) (prediction = 1, target = 0), true negatives (*TN*) (prediction = 0, target = 0), and false negatives (*FN*) (prediction = 0, target = 1) can be calculated. Finally, the *CSI*, *FAR*, and *HSS* metrics can be computed as follows:

$$CSI = \frac{TP}{TP + FN + FP} \tag{6}$$

$$FAR = \frac{FP}{TP + FP} \tag{7}$$

and

$$HSS = \frac{TP \times TN - FN \times FP}{(TP + FN)(FN + TN) + (TP + FP)(FP + TN)} \tag{8}$$

The selection of the threshold of 0.5 mm/h is consistent with the works by Shi et al. [41,42] and Xingjian et al. [43], and it can differentiate whether there is rain or not.

**4. Experiments**

*Precipitation Map Dataset*

We obtained three-dimensional radar from Swan radar from the Hubei Meteorological Service Center, which includes MCR, MTOP, MVIL, QPE30, QPE60, and MOSAIC maps at 6-min intervals from the last four years (2017–2020) in Hubei province. The dataset consists of approximately 1.8 million maps generated by 14 PD weather radars, with each raw map having dimension of 1200 × 800. During the period, 177 artificial rain-enhancement operations were conducted in Shiyan and 123 in Wuhan. The impact of artificial rainfall generally lasts for about 7 h, as shown in Figure 6. Therefore, to train the rainfall model

under natural conditions, we excluded the data influenced by artificial precipitation. We extracted a map with a dimension of 288 × 288, with the longitude and latitude of the evaluation site for artificial precipitation enhancement as the center, as the training data.

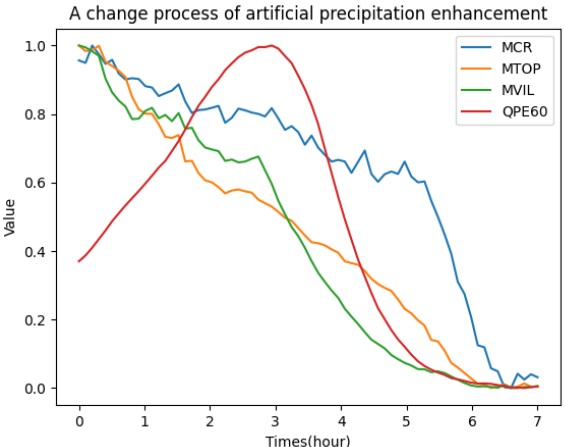

**Figure 6.** Duration of an artificial precipitation process.

To improve the model training, only data that meet the criteria for rainfall enhancement, as shown in Table 2, can be used as training data. One issue to address is determining the appropriate number of grid points for evaluating the rainfall enhancement criteria. Through the analysis of data from 2017 to 2020 (as shown in Figure 7), we found that selecting a 10 × 10 grid as the reference for rainfall enhancement criteria provides the most suitable data. As the grid size increases, the number of eligible training data decreases. Therefore, the optimal grid size is 10 × 10, which is consistent with the working area of most artificial precipitation enhancement operations.

**Table 2.** Rainfall enhancement criteria for stratiform and cumulus mixed clouds.

| Season | Altitude (Unit: m) | MCR (Unit: dbz) | MTOP (Unit: km) | MVIL (Unit: kg/m$^3$) |
|---|---|---|---|---|
| Spring (March to May) | <1000 | >20 | >4 | >5 |
| Summer (June to August) | <1000 | >25 | >5 | >10 |
| Autumn (September to November) | <1000 | >20 | >5 | >5 |
| Winter (December to February) | <1000 | >15 | >4 | >5 |

In the data preparation stage, we normalized the values in the training and testing sets by dividing them by the maximum value in the training set. Additionally, we cropped the precipitation maps to use only a subset of the original image (see Figure 8). This was because many pixels in the original image contained no data values, which was due to the maximum range of the radar being smaller than the image size (as shown in the black edges on the left panel of Figure 8). The rectangular area within the radar range is 1200 pixels × 800 pixels, equivalent to 1200 km by 800 km. For ease of neural network training, we performed a central crop of size 288 pixels × 288 pixels (as shown in the right panel of Figure 8). The crop size was determined based on the movement speed of clouds. On average, the speed of cloud movement is 36 km per hour, with a maximum of 50–60 km per hour. The speed and direction of clouds are influenced by factors such as wind speed, cloud height, and cloud density. In our algorithm, we selected 12 maps that met the requirements for cloud seeding as training inputs, with a time interval of 6 min between each map. If we do not consider the direction of movement, the maximum radius

of cloud movement in each map can be calculated as 72 km. Therefore, we chose a size of 144 pixels × 144 pixels for each map, which is sufficient for training. To ensure the generalizability of the training, we doubled the length and width of each map.

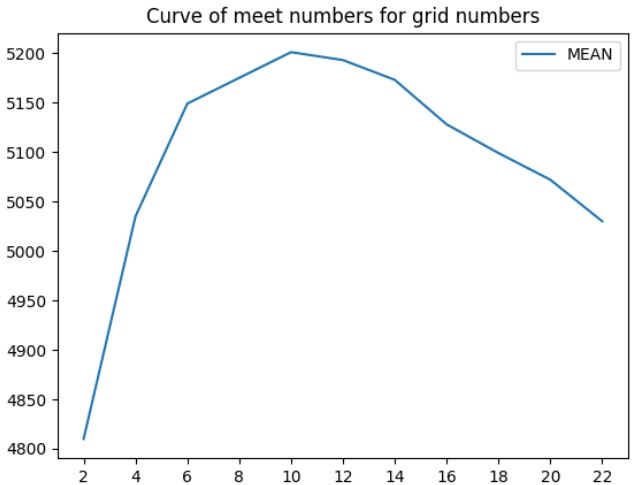

**Figure 7.** The relationship between grid size selection and the amount of data eligible for rainfall enhancement.

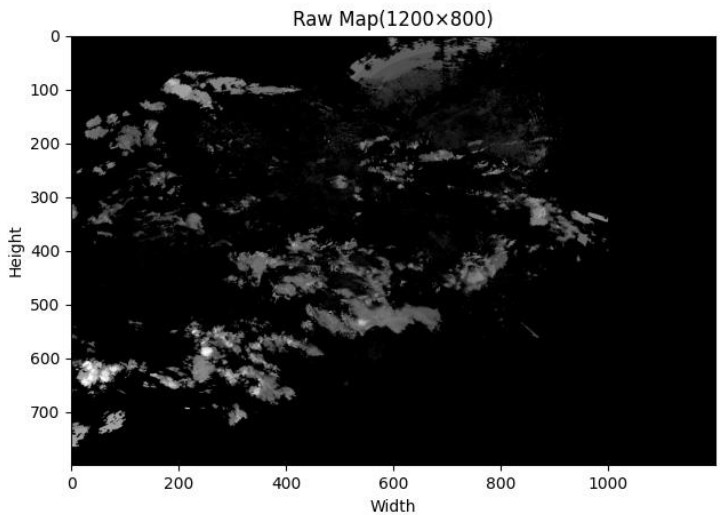
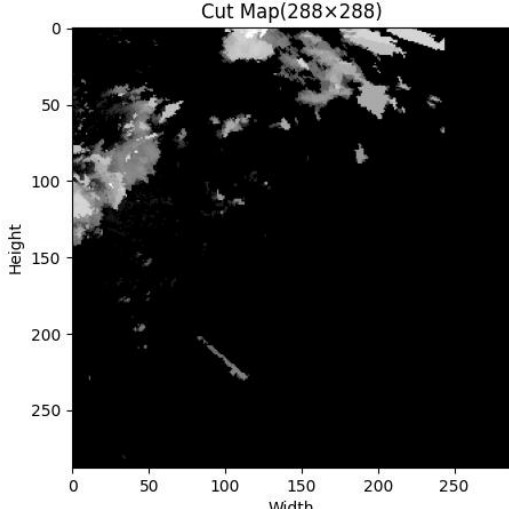

**Figure 8.** Weather radar data map.

The output of the training is the rainfall of each grid point in each region. Actual grid rainfall data is actually difficult to obtain. We usually only obtain 1-h resolution rainfall data by radar retrieved quantitative precipitation estimation (QPE) and discrete monitoring station rainfall data for each region. Therefore, these two 1-h resolution data must be converted into 6-min resolution grid data for algorithm training.

At present, radar measurement of precipitation is mainly based on *Z-I* relationship [44–46]. That is $Z = A \times I^b$, where $Z$ is the radar reflection factor (unit: $\mathrm{mm}^3/\mathrm{m}^6$). I is rainfall intensity (unit: mm/h). *A* and *b* are coefficients. The accuracy of quantitative precipitation estimation depends to a large extent on the determination of *A* and *b* parameters in the *Z-I* relationship. Because the precipitation properties are different in different seasons and locations, the *Z-I* relationship is also different. At present, many stations still only use the fixed *Z-I* relationship provided by the manufacturer to estimate ground precipitation. With the construction of a large number of encrypted automatic weather stations, the spatial and temporal density of precipitation observation has greatly increased. It has become a reality to make full use of the encrypted ground precipitation observation data and the intensity

of radar echo to carry out high-precision *Z-I* relationship analysis. Many domestic scholars have also carried out relevant research. This paper proposes a specific technical scheme based on this problem. By integrating the radar network mosaic data of daily business applications and the precipitation observation data of the ground encrypted automatic weather station, based on the optimization method, the local dynamic *Z-I* relationship is established, and the quantitative precipitation inversion data with 6-min resolution is obtained in real time. The optimization algorithm is divided into the following three steps:

(1) Based on the rainfall *Z-I* relationship, convert the 6-min radar real reflectivity factor *Z* in the past hour into the radar-estimated rainfall *I*, and then accumulate the 6-min radar-estimated rainfall *I* to obtain the hourly radar-estimated rainfall, so as to compare it with the precipitation observed by automatic ground stations.

(2) In order to obtain the optimal parameters *A* and *b* for radar retrieval of precipitation, the hourly radar-estimated precipitation is *R* and the ground automatic station observed precipitation is *G*, and the error target discriminant function *CTF* is selected:

$$CTF = \min\left\{ \sum_{i=1}^{n} \left[ (R_i - G_i)^2 + |R_i - G_i| \right] \right\} \tag{9}$$

In Equation (9), *R* is the hourly radar-estimated precipitation; *G* is the precipitation observed by the automatic ground station; *n* is the total logarithm of radar automatic station data matching involved in rainfall Z-I relationship fitting. In practical business applications, in order to save calculation time and ensure that parameters *A* and *b* change within a reasonable range, the adjustment ranges of *A* and *b* are limited to [150.00, 400.00] and [0.80, 2.40] respectively, and the adjustment intervals are 0.10 and 0.01 respectively. For each group of *A* and *b*, a *CTF* can be obtained. By constantly adjusting the combination of *A* and *b*, it is determined that the *Z-I* relationship of precipitation determined by Equation (9) A and b whose error objective discriminant function *CTF* reaches the minimum is optimal.

(3) Convert the precipitation *Z-I* relationship obtained in step (2) of the 6-min radar reflectivity factor prediction field within the current 1 hour into precipitation, and then accumulate it into hourly radar quantitative precipitation retrieval data to meet the needs of precipitation inspection.

In addition, in order to quantitatively analyze the precipitation inversion error of the dynamic *Z-I* relationship method, the mean error ($E_{ME}$), mean relative error ($E_{MRE}$) and other test parameters are calculated.

The error calculation equation is:

$$E_{ME} = \frac{1}{n} \sum_{i=1}^{n} (R_i - G_i) \tag{10}$$

$$E_{MRE} = \frac{1}{n} \sum_{i=1}^{n} \frac{|R_i - G_i|}{G_i} \tag{11}$$

Equations (10) and (11), $R_i$ and $G_i$ are the precipitation inversion value and real value of the automatic ground weather station respectively; *n* is the total number of sites.

## 5. Results and Discussion

After training the three discussed models, we selected the model with the lowest validation loss for each model. These best-performing models were then used to calculate several metrics introduced in Section 3 on the test set. The models were trained, evaluated, and tested on the dataset. In precipitation nowcasting, a common baseline is the persistence method. The persistence model predicts the last input image of a sequence as the prediction image, based on the assumption that the weather will not significantly change from time point *t* to *t* + 1. Especially in nowcasting, this baseline is not easy to outperform because

the time differences between images are so short (e.g., 2 or 6 min) that weather conditions often remain the same [47].

## 5.1. Evaluation on Precipitation Map Dataset

The results obtained on the dataset are presented in Table 3. It is important to note that the *MSE* values were calculated after denormalizing the model predictions to the original rainfall (mm/6 min). The results demonstrate that every model we tested outperformed the common persistence baseline by a significant margin on the Precipitation map dataset. This is remarkable since, as mentioned earlier, it can be challenging to surpass this baseline in nowcasting due to the short time differences between the input and target.

**Table 3.** *MSE* and scores on rainfall bigger than 0.5 mm/h indicating rain or no rain. Best result for that score is in bold. A ↑ indicates that higher values for that score are good whereas a ↓ indicates that lower scores are better.

| Model | *MSE* ↓ | *Accuracy* ↑ | *Precision* ↑ | *Recall* ↑ | *F1* ↑ | *CSI* ↑ | *FAR* ↓ | *HSS* ↑ |
|---|---|---|---|---|---|---|---|---|
| Persistence (baseline) | 1.1697 | **0.7264** | 0.7315 | 0.8313 | **0.729** | **0.5735** | **0.2736** | 0.4039 |
| UNet | 0.1239 | 0.6615 | 0.8530 | 0.7913 | 0.5078 | 0.3403 | 0.3385 | 0.3951 |
| CoGRU | 0.1542 | 0.6294 | 0.6643 | 0.8042 | 0.5216 | 0.3529 | 0.3706 | **0.4238** |
| UNet-GRU | **0.1182** | 0.6311 | **0.874** | **0.8462** | 0.5192 | 0.3506 | 0.3689 | 0.4139 |

Based on the figure, it is evident that the UNET-GRU model outperforms other models in terms of capturing the development of heavy rain clusters and accurately describing the vertical distribution of rain clusters. However, the hybrid model, which combines the strengths of multiple models, is even more superior in performance compared to any single model.

## 5.2. Evaluate the Effect of Rainfall Enhancement

We conducted a 7-h rainfall inversion based on two artificial rainfall cases implemented in Shiyan and Wuhan, both in Hubei Province. Information regarding the rain enhancement operation is presented in Table 4. The inversion results were compared with the actual rainfall, as shown in Figures 9 and 10. The results indicated that the inversion accuracy in Shiyan was higher compared to that in Wuhan. This may be attributed to the fact that Shiyan is situated in a mountainous region and is less influenced by human activities. The radar data are relatively stable, leading to more accurate predictions.

**Table 4.** Description of information related to artificial precipitation enhancement.

| No. | Date | Rockets (pcs) | Start Time | End Time | Conditions before op. | Conditions after op. | Area (km$^2$) | Effect | Region |
|---|---|---|---|---|---|---|---|---|---|
| 1 | 30 July 2017 | 6 | 05:58:10 | 06:52:40 | Light to moderate rain | Moderate to heavy rain | 400 | good | Wuhan |
| 2 | 26 April 2018 | 4 | 00:06:32 | 00:48:22 | overcast | light rain | 360 | good | Shiyan |

Data from the rainfall monitoring station and the rain enhancement operation record file (Table 4) show that the meteorological conditions in Wuhan before artificial rain operation were light to moderate rain (0.20 mm/h to 0.7 mm/h), consistent with the rainfall predicted through deep learning in Figure 9a. Based on the natural catalytic inversion lasting for 7 h, starting from 5:58:10 a.m. on July 30, 2017, it can be inferred that if the artificial rain operation had not been carried out, the rainfall during the 7-h period would have been only 3.56 mm, whereas the actual rainfall was 18.91 mm, with a residual value of

15.35 mm (Table 5). This indicates a significant rain enhancement effect, which is consistent with the results recorded in historical files for this artificial rain operation (Table 4).

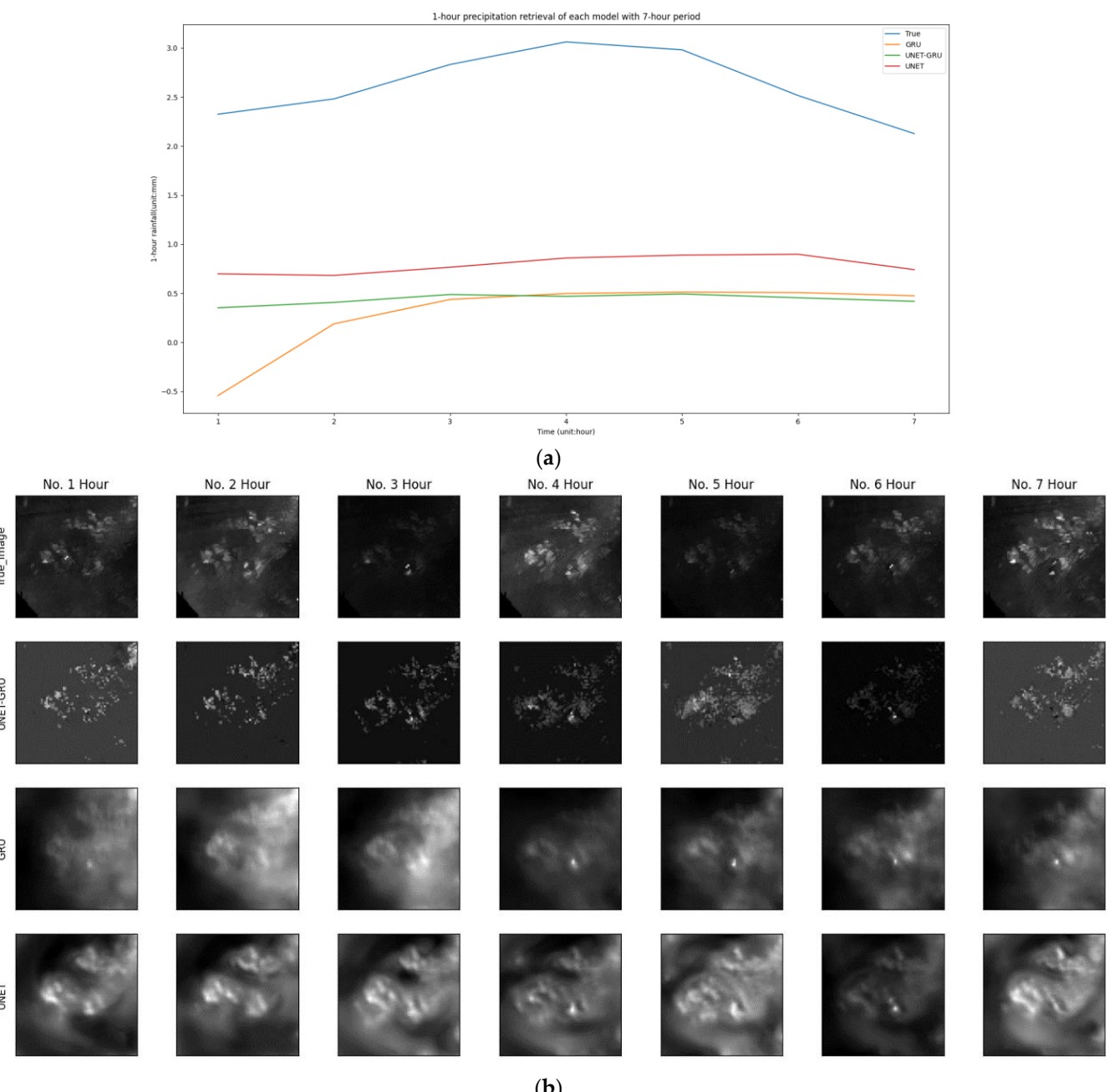

(**a**)

(**b**)

**Figure 9.** Comparison of 7-h grid and average rainfall inversion for a simulated artificial rainfall event in Wuhan region. (**a**) Comparison chart of 1-h rainfall forecast and actual measurement values for each model (lasts for 7 h). (**b**) Comparison of 1-h grid-based rainfall forecasts and observed values from different models (lasts for 7 h).

**Table 5.** Effect estimates of artificial precipitation compared with historical records.

| No. | Date | Start Time | Duration | Naturally Evolved Rainfall | Actual Rainfall | Residual Rainfall | Effect | Region |
|-----|------|------------|----------|----------------------------|-----------------|-------------------|--------|--------|
| 1 | 30 July 2017 | 05:58:10 | 7 h | 3.56 mm | 18.91 mm | 15.35 mm | good | Wuhan |
| 2 | 26 April 2018 | 00:06:32 | 7 h | 1.05 mm | 11.03 mm | 9.98 mm | good | Shiyan |

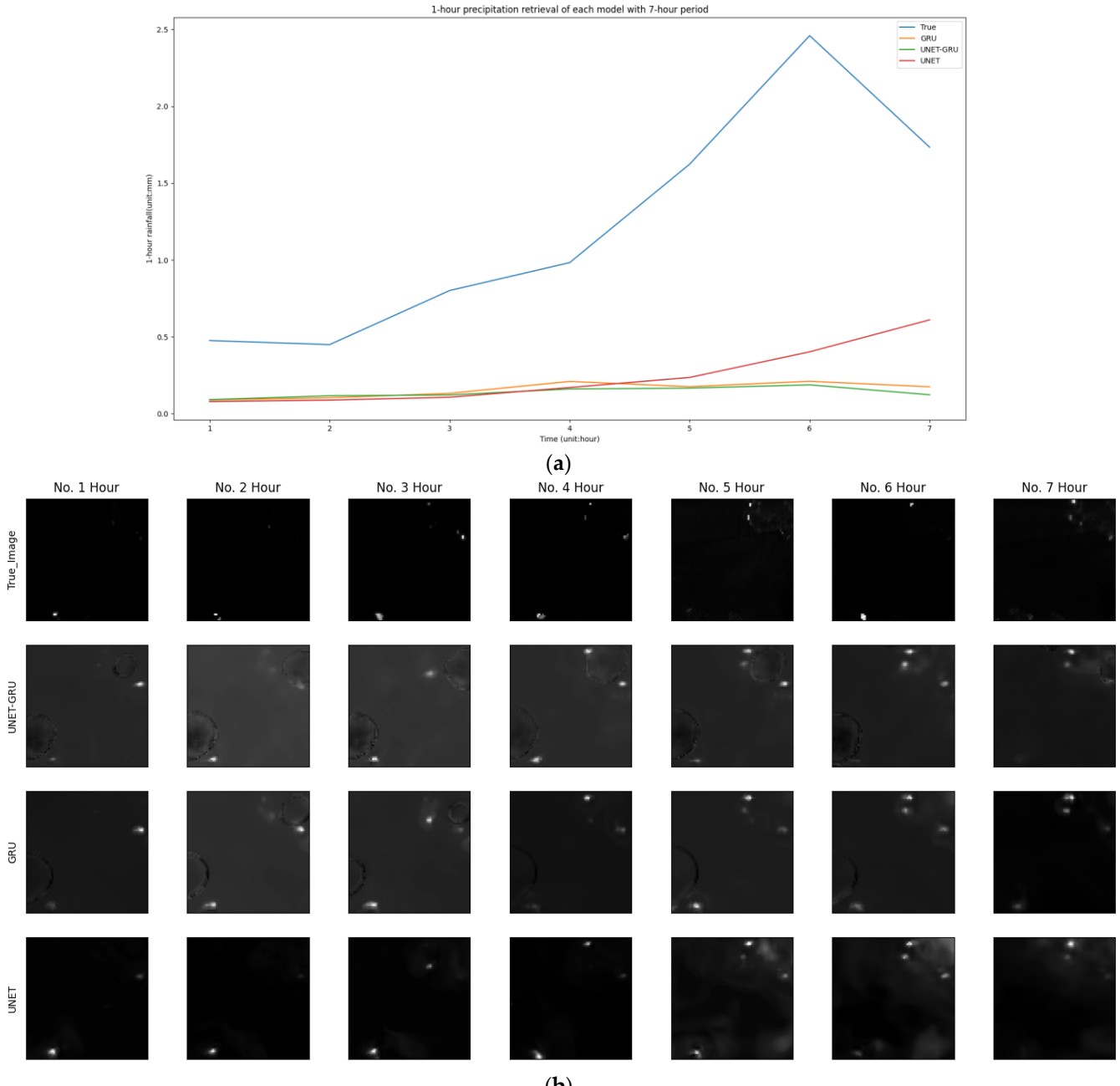

**Figure 10.** Comparison of 7-h grid and average rainfall inversion for a simulated artificial rainfall event in the Shiyan region. (**a**) Comparison chart of 1-h rainfall forecast and actual measurement values for each model (lasts for 7 h). (**b**) Comparison of 1-h grid-based rainfall forecasts and observed values from different models (lasts for 7 h).

Similarly, the artificial rain operation carried out in Shiyan, China on 26 April 2018 at 12:32 a.m. also shows good results. Before the operation, the weather was cloudy and the rainfall changed only slightly. However, the rainfall began to increase in the fourth hour and reached its peak in the sixth hour. The artificial rain operation lasted for 7 h and produced a rainfall of 11.03 mm, while the natural evolution only produced 1.05 mm of rainfall. This fully demonstrates the success of this artificial rain operation.

From Figures 9a and 10a, it can be observed that in the rainfall estimation algorithm, the rainfall estimated by the UNET algorithm is generally larger than that estimated by the UNET-GRU and GRU algorithms. The consistency of rainfall estimation by the UNET-GRU and GRU algorithms is relatively good, but in Figure 9a, the GRU algorithm shows

prediction distortion in the first hour, indicating that the robustness of the GRU algorithm is inferior to that of the UNET-GRU algorithm. From Figures 9b and 10b, it can also be observed that the UNET-GRU algorithm is superior to the other two algorithms. The rainfall maps predicted by the GRU and UNET algorithms are blurred, while the rainfall map predicted by the UNET-GRU algorithm has relatively high clarity.

## 6. Conclusions

In this study, we propose a novel evaluation method to assess the effectiveness of artificial precipitation enhancement. Our approach involves training a weather evolution and development model using artificial neural networks, which simulates the natural catalysis process under increasing rainfall conditions. This eliminates the need for a traditional evaluation method that requires selecting a contrast test area. Our experimental results demonstrate that the effect of artificial precipitation enhancement can be quantitatively evaluated by a grid, rather than relying solely on qualitative assessment, thereby making the evaluation method more reliable and scientifically sound. To demonstrate the universality of our approach, we tested our evaluation method using two sets of artificial rainfall data for 7-h rainfall inversion.

In some cases, the SWAN radar data used for algorithm training may be contaminated due to terrain effects, especially in densely populated urban areas, which can lead to an increase in prediction errors. Therefore, it is necessary to clean the radar data before algorithm training. Additionally, the model currently has approximately 21.6 million parameters, and training once on one piece of RTX3090 graphics card takes about 10 hours, which is unacceptable. Therefore, optimizing the model is an aspect worth researching.

**Author Contributions:** Conceptualization, R.L.; Data curation, D.L. and H.Z.; Formal analysis, R.L. and D.L.; Funding acquisition, L.Z.; Investigation, L.Z. and P.X.; Methodology, R.L., D.L. and P.X.; Project administration, R.L.; Software, R.L. and P.X.; Supervision, L.Z. and H.Z.; Writing—original draft, R.L.; Writing—review and editing, R.L. All authors have read and agreed to the published version of the manuscript.

**Funding:** This work was supported by Hubei Provincial Natural Science Foundation of China (Grant No. 2022CFD019) and Open project of Hubei Provincial Key Laboratory of Intelligent Robot, the Innovation and Development Project of China Meteorological Administration (Grant No. CXFZ2022J036) and the Wuhan Knowledge Innovation Special Project (Grant No. 2022022101015009) and the National Natural Science Foundation of China (Grant No. 62171327).

**Data Availability Statement:** The data used to support the findings of this study are available from the corresponding author upon request. Due to the fact that meteorological radar data is classified as national confidential data, the data has not been made public.

**Conflicts of Interest:** The authors declare that they have no conflict of interest.

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
