# Peer review of "Evaluation of Artificial Precipitation Enhancement Using UNET-GRU Algorithm for Rainfall Estimation"

_water, doi:10.3390/w15081585_

Round 1
Reviewer 1 Report
As in the report

Reviewer 2 Report
Dear Editor,
Water
In this study, we propose a novel evaluation method to assess the effectiveness of artificial precipitation enhancement. The subject addressed is interesting and within the scope of the Water. Nevertheless, some major revisions have been found:
- The quantified results must be added to the abstract.
- The literature review is too old, and the new and applicable studies should be reviewed. You can be referenced the following articles in the introduction:
https://doi.org/10.2166/wcc.2022.066
- doi:10.3808/jei.202200473
- In the last paragraph of the introduction, first, the disadvantages of previous studies should be stated, then based on these disadvantages, the innovation of the present study should be mentioned. I suggest adding a new section on study innovation.
- Please add a new section to Method for presenting statistical criteria of inputs data.
- MSE and NMSE are not presented in Table 4.
- The caption of Table 4 and Table 3 are similar to each other.
- The results of the present study should be compared with previous studies.
- The results and discussion section is too short. More discussion should be added to this section. Why does UNet-GRU perform better than other algorithms? Please explain this issue.
- In the conclusions, a summary of the work method and the most important achievements and results should be presented. The limitation of the employed approach for predicting precipitation and temperature should be added. Please add recommendations for future studies.
Considering the mentioned points, this study in the current version needs major revisions.
Reviewer 3 Report
1. What is the main question addressed by the research?
The paper is dedicated to evaluation of the effects of artificial precipitation enhancement. Authors propose a deep learning-based method for quantitative evaluation of the effect of artificial rainfall. The obtained results demonstrate that their method improves the accuracy of the effect evaluation and enhances the ability of the evaluation scheme.
2. Do you consider the topic original or relevant in the field, and if so, why?
The topic is original and relevant in the field. Authors propose an evaluation method to assess the effectiveness of artificial precipitation enhancement. Their approach involves training a weather evolution and development model using artificial neural networks, which simulates the natural catalysis process under increasing rainfall conditions. The experimental results demonstrate that the effect of artificial precipitation enhancement can be quantitatively evaluated by a grid, rather than relying solely on qualitative assessment, thereby making the evaluation method more reliable and scientifically sound.
3. What does it add to the subject area compared with other published material?
Authors propose a deep learning-based method for quantitative evaluation of the effect of artificial rainfall. Their approach involves training a weather evolution and development model using artificial neural networks, which simulates the natural catalysis process under increasing rainfall conditions. The universality of the approach has been proven by tests of the evaluation method with two sets of artificial rainfall data for 7-hour rainfall inversion.
4. What specific improvements could the authors consider regarding the methodology?
There is no need to make any improvements or something else. The authors outlined the current achievements in the field in the introduction, and provided detailed description of the materials and method, as well as the obtained results, and conclusions. The results obtained are based on experimental evaluation of their approach. However, authors need to provide more detailed discussion of the results obtained.
5. Are the conclusions consistent with the evidence and arguments presented and do they address the main question posed?
The conclusions are consistent with the evidence and arguments presented in the manuscript and address the main questions of their study. However, it should be extended and its part with Figures 9 and 10 should be moved to the section Results and Discussion. Also, the future research perspective should be outlined in Conclusions.
6. Are the references appropriate?
The references are appropriate, and most of them are fairly up-to-date. Nevertheless, authors should prepare the References section exactly in accordance with the journal template, abbreviate names of journals, and provide missing DOIs.
7. Please include any additional comments on the tables and figures.
All the tables and figures are appropriate. They show well the research and experiment details and results.
Figure 1. All the arrow lines should connect to the appropriate blocks.
Table 4. Columns with ‘Conditions’ should be widening so as avoid overlapping these words.
Please, enlarge text size in Figures 9 and 10. Remove one of two (a) in Figure 9a caption.
8. Other comments.
Please, use ‘Equation ()’ instead of ‘formula ()’.
Author Contributions paragraph is missing. Please, provide it in accordance with the journal template.
After detailed consideration of the manuscript, I have found that the results obtained are new and significant for the field. The manuscript is written well but needs some corrections before its publication in the journal.
So, the paper needs a minor revision.
Round 2
Reviewer 2 Report
Dear Editor,
Water
This manuscript is previously been carefully evaluated. The current version of the manuscript is acceptable.